# Revealing 3D structure of gluten in wheat dough by optical clearing imaging

Takenobu Ogawa [1✉] & Yasuki Matsumura [2✉]

Gluten, which makes up 85% of endosperm wheat protein, is considered a crucial quality determinant of wheat-based food products. During wheat dough manufacture, the molecular packing of gluten causes formation of large structures that exceed the millimetre scale. However, due to lack of imaging techniques for complex systems composed of giant macromolecules, the entire gluten structure remains unknown. Here, we develop an optical clearing reagent (termed SoROCS) that makes wheat-based products transparent. Combined with two-photon microscopy, we image the three-dimensional (3D) structure of gluten at the size in the millimetre scale and at submicron resolution. Further, we demonstrate how the 3D structure of gluten dramatically changes from a honeycomb-shaped network to sparse large clumps in wheat noodles, depending on the salt added during dough making, thereby reducing stress when compressing the noodle. Moreover, we show that SoROCS can be used for noodle imaging using confocal laser scanning microscopy.

[1] Division of Food Science and Biotechnology, Graduate School of Agriculture, Kyoto University, Kyoto, Japan. [2] Division of Agronomy and Horticultural Science, Graduate School of Agriculture, Kyoto University, Uji, Kyoto, Japan. ✉email: ogawa.takenobu.6v@kyoto-u.ac.jp; matsumur@kais.kyoto-u.ac.jp

Gluten is a storage protein found in certain cereal grains and accounts for 85% of the endosperm protein in wheat[1]. Gluten contains high levels of glutamine, nonpolar amino acid proline and glycine and low levels of amino acids with ionisable side chains; hence, it has the potential to exhibit extensive polymerisation behaviour in liquid solutions[2]. Because polymerisation by the molecular packing of gluten causes the formation of large structures that exceed the millimetre scale during wheat dough manufacture[3,4], gluten plays a key role as a skeleton in wheat dough. These distinctive features of gluten provide wheat dough with a unique viscoelasticity[5], which allows us to produce various wheat-based foods and determines the intermediate and final quality of gluten-containing foods[6]. However, poor solubility and a lack of crystallinity have been major barriers in determining its structure[7]. Therefore, previous studies have only provided images limited to two dimensions (2D) by generating mechanical sections. Although a myriad of microscopic techniques have been employed to observe gluten, including confocal fluorescence microscopy[8,9], confocal Raman microscopy[10], scanning electron microscopy[11] and transmission electron microscopy[12], to our knowledge, no 3D structure has been obtained. Therefore, despite the numerous scientific studies that have been conducted on gluten over the past three centuries since it became the first protein to be separated from a natural source excluding animal products[5], we still have a minimal understanding of the overall gluten structure.

A significant breakthrough has been made in light microscopy-based connectomics of cellular 3D structures[13–18]. The current pioneering method to view 3D structures in biological samples, such as mammal brains, involves mechanical sectioning and reconstruction. Briefly, a sample is continuously sliced with a microtome, and 2D images of the mechanically sectioned sample are taken with a microscope. A set of continuous 2D images is reconstructed in 3D using a PC software. Although this method enables access to deeper structures, it requires sophisticated techniques and labour and causes the destruction of microstructures in some cases. In contrast, optical sectioning allows simple and fast imaging for 3D reconstruction of fluorescently labelled structures in samples, combined with two-photon excitation microscopy (2PEM) or light-sheet microscopy. An obstacle facing deep imaging with optical sectioning is light scattering, which is caused by a mismatch in the refractive indices (RI) between the components and surrounding solvents. Several optical clearing reagents have been developed to reduce the amount of light scattering in biological samples, such as mammal brains and plants[13–19]. They aim to remove or denature the high-RI components, such as proteins, lipids and chlorophyll, and to replace the solvent with a high-RI reagent.

Meanwhile, there are no published studies that address the optical clearing of samples including wheat-based products. Wheat flour contains 80–85% starch, 8–14% proteins[20] and 1% lipids, suggesting that starch will be a major source of light scattering in wheat-based samples. Therefore, the removal of starch and/or adjustment of the RI difference between starch and the surrounding substances is a potential approach to increase the transparency of wheat-based samples. However, removal of starch would inevitably damage wheat-based samples with profound loss of structural information on gluten since starch is present in wheat flour as large particles, ~30 μm in size, rather than small molecules. Therefore, adjusting the RI of wheat-based products has been a challenge for us.

In this work, we develop an optical clearing reagent (termed sodium salicylate-based reagent optically clears starchy-products (SoROCS)) that makes wheat-based products transparent within a few days. Combined with two-photon microscopy, we image the 3D structure of gluten at the size in the millimetre scale and at submicron resolution.

## Results

**A reagent for optical clearing.** The development of a reagent for optical clearing was based on a serendipitous discovery. While investigating the effect of various types of salt, contained in the boiled solution, on the water sorption kinetics of noodles, we found that noodles immersed in a solution with a very high concentration of sodium salicylate became transparent. Sodium salicylate is a known analgesic and antipyretic in medicine that acts as a non-steroidal anti-inflammatory drug. The mechanism by which sodium salicylate causes the noodles to become transparent is not fully understood, however, it may be partially explained by the gelatinisation of starch. Gelatinisation is a phenomenon whereby the starch granules begin to swell with a gradual increase in the transparency and viscosity when starch is heated in the presence of water. Salt is one of the substances that affect gelatinisation, and the order in which salt promotes gelatinisation follows the Hofmeister series[21]. The order of the Hofmeister series for anions and cations is $OH^- > C_6H_4(OH)COO^- > SCN^- > I^- > Br^- > Cl^- > SO_4^{2-}$ and $Ba^{2+} > Ca^{2+} > Mg^{2+} > Li^+ > Na^+ > K^+ > Rb^+ > NH_4^+$. Hence, sodium salicylate is a relatively strong gelatinisation promoter. Indeed, sodium hydroxide is a candidate for further promotion of gelatinisation, however, its high pH makes it unsuitable for use as an optical clearing reagent: alkalies induce protein denaturation[22]. Furthermore, 3D imaging by optical sectioning requires fluorescent labelling of structures, however, high pH solutions may induce fading fluorescence of the labelling reagents.

This serendipitous discovery inspired us to develop an optical clearing reagent for wheat-based products. Since the solubility of sodium salicylate in water at 20 °C is ~4.125 M, we set its concentration to 4 M in the current study. Moreover, referring to the previous studies that addressed clearing reagents for biological samples[13], we added Triton X-100 as a detergent for lipids at a concentration of 0.1% (wt wt$^{-1}$) to 4 M sodium salicylate solution. This solution, which we designated SoROCS, has a pH of 7.5 and a high RI of 1.455 at room temperature and is almost colourless (Supplementary Fig. 1).

**Transparency assessment.** To examine the performance of SoROCS, we prepared wheat noodles, fixed them with 4% paraformaldehyde for 1.5 h and immersed them in water (0.5% (wt wt$^{-1}$) sodium azide added to prevent spoilage), ClearSee or SoROCS (Fig. 1a–h). ClearSee, which is reportedly able to make plants transparent, was previously developed based on the screening of 24 compounds, including polyhydric alcohols, surfactants and urea, with the aim of removing autofluorescence of chlorophyll from green leaves[19]. Herein, to evaluate the transparency of noodles, we quantified the opacity of the transmitted light image. When the noodles were fixed with 4% paraformaldehyde for 1.5 h, the opacity was observed to increase from $3.54 \pm 0.03$ to $5.58 \pm 0.05$ (Fig. 1i); after which the noodles immersed in water remained opaque, whereas the noodles treated with ClearSee or SoROCS began to become transparent within 6 h (Fig. 1a–h). The opacity of the noodles immersed in ClearSee and SoROCS for 3 days was $1.61 \pm 0.04$ and $1.27 \pm 0.01$, respectively (Fig. 1i). To further compare the effect of treatment with ClearSee and SoROCS on the transparency of noodles, we laid a grid-printed film under the noodles and examined how clearly the grid pattern penetrated the noodles (Fig. 1j–o). Compared to the noodles treated with ClearSee, the plot profile for those treated with SoROCS demonstrated a large difference in grey level between the line and the non-line sections of the grid, and the peaks of the grey

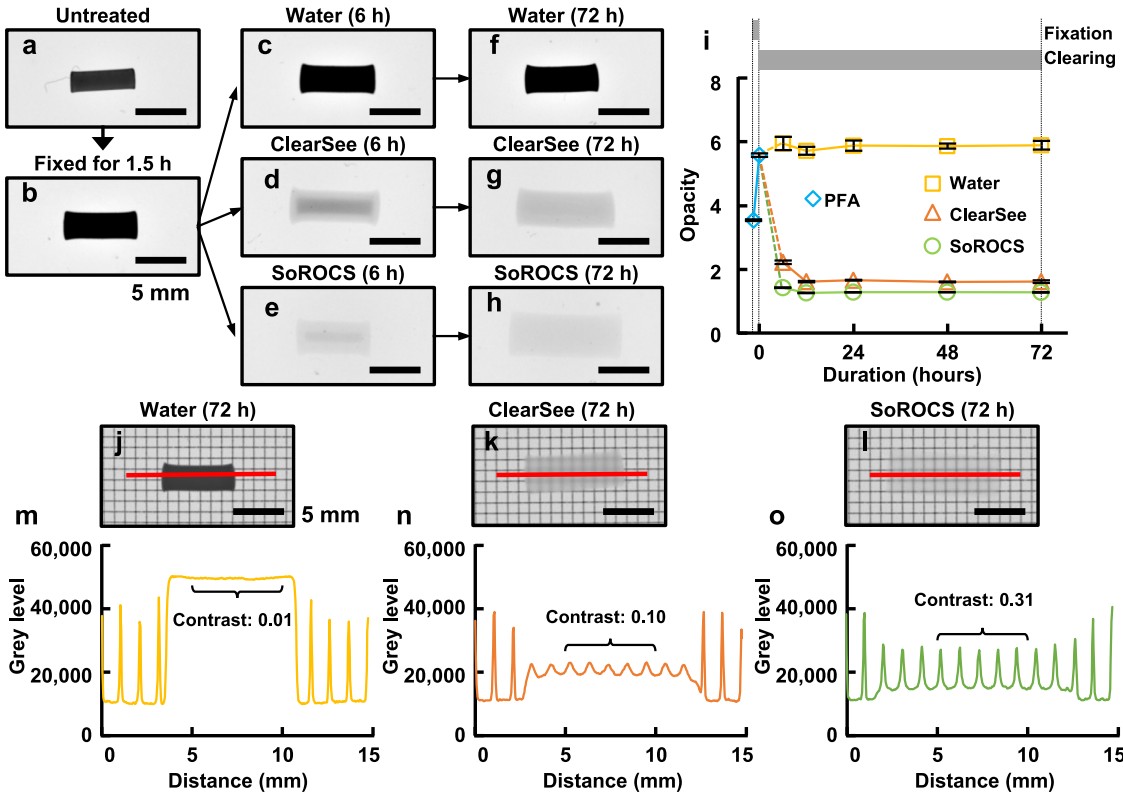

**Fig. 1 Optical clearing performance of SoROCS. a–h, j–l** Photographs were obtained using ImageQuant for transmission. Note, none of the samples were the same and they were prepared separately (**a–h**); however, the images presented in (**j–l**) are the same samples as in (**f–h**), respectively, with a grid pattern-printed film as a background. Scale bars indicate 5 mm (**a–h, j–l**). Untreated noodle sample (**a**) was fixed with 4% PFA/PBS for 1.5 h (**b**) and immersed in water (0.5% (wt wt$^{-1}$) sodium azide added to prevent spoilage) (**c, f**), ClearSee (**d, g**) or SoROCS (**e, h**). **i** Opacity curves in the fixation process (light blue) and clearing process using water (yellow), ClearSee (orange) or SoROCS (green) were calculated from the transmission images of noodles without the grid pattern-printed film. Data are presented as mean values ± SD ($N = 3$ independently prepared noodle samples). Plot profile for the noodle treated with water (**m**), ClearSee (**n**) or SoROCS (**o**). The red line in (**j–l**) indicates the corresponding location of the plot profile in (**m–o**), respectively. The contrast value in the plot profile for the areas shown in curly brackets is shown in (**m–o**), respectively. Source data underlying (**i, m–o**) are provided as a Source Data file.

level that reflect the lines were sharp. The contrast $C$ represents the difference in grey level and was calculated according to Eq. (1). The resulting $C$ for ClearSee and SoROCS values in a 5-mm long plot profile (curly braces in Fig. 1n, o) was 0.10 and 0.31, respectively. The high contrast indicates that SoROCS can further suppress light scattering, which is a common obstacle associated with deep imaging via optical sectioning. Of note, the noodles treated with ClearSee had visible crevices, as indicated by the red arrows in Supplementary Fig. 2a, implying that the noodle structure was destroyed. A previous paper mentioned that mouse brains became very fragile and unwieldy when treated with a clearing reagent containing urea because urea causes partial denaturation and loss of cellular proteins[16]. In contrast, the noodles treated with SoROCS, which does not contain urea, did not show marked fragility (Supplementary Fig. 2b).

**Expansion assessment.** Considering that previously reported incipient optical clearing reagents for biological samples caused large changes in sample volume[16], we also investigated the morphological changes resulting from optical clearing. Linear expansion was observed to increase for noodles immersed in water and SoROCS (Fig. 2a). Specifically, the linear expansion suddenly increased in the early stage, and subsequently gradually approached the maximum value. The linear expansion values following immersion in water and SoROCS for 3 days were 1.28

± 0.01 and 2.23 ± 0.08, respectively. Together, these results suggest that the increase in opacity of the noodles immersed in water is due to the expansion of noodles. In contrast, the noodles immersed in SoROCS swelled considerably, however, they increased their transparency. This considerable expansion of SoROCS-treated samples may be due to the swelling of starch induced by sodium salicylate.

Next, to examine the effect of expansion on gluten structures, we conducted protein network analysis, which is an image analysis that allows for precise quantification of structural and morphological network attributes of the complex gluten structure[23–25], for non-cleared noodles and cleared noodles. First, we prepared noodles fixed with 4% paraformaldehyde for 1.5 h and captured a 2D image of gluten in the sample section that was mechanically sliced using a cryostat (Fig. 2b). Gluten was fluorescently labelled with Thiolite$^{TM}$ Green (TG), which reacts with thiol groups (cysteine residues) on proteins (see Supplementary Fig. 3 for the gluten staining specificity by TG). Next, we cleared the noodle with TG-mixed SoROCS (the noodle was fixed with 4% paraformaldehyde for 1.5 h before clearing) and visualised the gluten by optical sectioning (Fig. 2c). Note, the mechanically sectioned images of non-cleared noodles (Fig. 2b) and the optically sectioned images of cleared noodles (Fig. 2c) were observed with ×20 and ×10 objective lenses, respectively, since the linear expansion of the noodles fixed with 4% paraformaldehyde for 1.5 h, and noodles cleared with SoROCS for 3 days after fixation

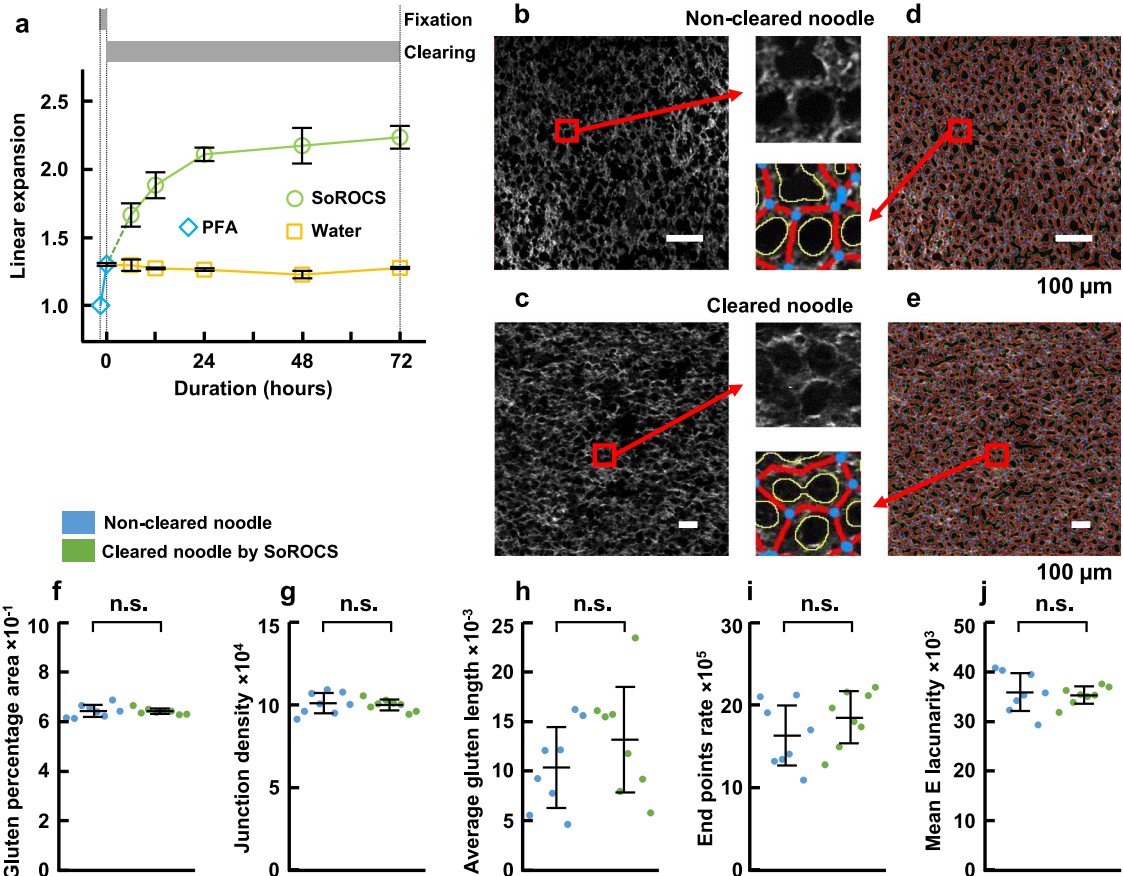

**Fig. 2 Expansion characterisation during clearing. a** Linear expansion curves in the fixation process (light blue) and clearing process using water (yellow) or SoROCS (green). Data are presented as mean values ± SD (N = 3 independently prepared noodle samples). **b** CLSM image of gluten in the non-cleared noodle sample section, which was mechanically sliced using a cryostat. **c** 2PEM image of gluten in the cleared noodle by SoROCS. **d, e** Image after protein network analysis with AngioTool (blue points: junction (branching points), red lines: gluten skeleton, yellow line: gluten outlines). An enlarged image of the section delineated with a red square is shown at the tip of the red arrow in (**b–e**). Scale bars indicate 100 μm (**b–e**). Comparison of variables calculated by AngioTool for non-cleared noodles (blue) and cleared noodles (green): gluten percentage area (**f**), junction density (**g**), average gluten length (**h**), end points rate (**i**) and mean E lacunarity (**j**). Noodle samples were prepared four times independently (N = 4), and two different section samples were prepared from each noodle preparation (N = 2). One of these eight different images obtained using CLSM and 2PEM is shown in (**b, c**), respectively. **f–j** Date are presented as mean values ± SEM. The SEM is calculated from the mean values for N = 2. P values were calculated using the Student's two-tailed unpaired t test (**f**: P = 0.92, **g**: P = 0.76, **h**: P = 0.38, **i**: P = 0.31, **j**: P = 0.76). All variables show no significant difference at the P = 0.05 level. Source data underlying (**a, f–j**) are provided as a Source Data file.

was 1.30 ± 0.01 and 2.23 ± 0.08, respectively (Fig. 2a). We used the AngioTool to conduct protein network analysis on both images (Fig. 2d, e). AngioTool was originally developed for the quantitative analysis of angiogenesis[26], however, considering that the gluten network shares many commonalities with blood vessels, it has been successfully applied for the accurate quantification of structural and morphological network attributes of gluten structures[23–25]. By adjusting the parameters used in AngioTool with the skeletal structures of gluten (Fig. 2b–e), we obtained variables that characterise the morphology of gluten structure: gluten percentage area, junction density, average gluten length, end points rate and mean E lacunarity (Supplementary Table 1). No significant differences were observed in variables between the mechanical sectioning images of non-cleared noodles and the optical sectioning images of cleared noodles (P > 0.05 for each; Fig. 2f–j). Although the expansion caused by the swelling of starch granules during the clearing process using SoROCS may be an artefact of deep imaging, we, nevertheless, determined that expansion does not significantly impact the skeletal structures of gluten in terms of morphology.

**Fluorescence imaging of gluten**. To image with the 2PEM, we first sought to explore the optimal wavelength for two-photon excitation. We prepared the noodle cleared with TG-mixed or Alexa Fluor™ (AF) 633 C5 Maleimide-mixed SoROCS and measured fluorescence intensity by changing the excitation wavelength from 920 to 960 nm (Fig. 3a–f) or from 820 to 860 nm (Fig. 3g–l), respectively, at 10-nm intervals. The fluorescence intensity histogram of the obtained images shows a slightly higher count in the region of high fluorescence intensity between ~500 and 1000 when excited at 940 nm for TG staining and between ~1500 and 2000 when excited at 840 nm for AF staining (Fig. 3f, l).

Next, to examine the image resolution for SoROCS-treated samples obtained with 2PEM, we prepared noodles kneaded with fluorescent microspheres (0.49 μm diameter), and subsequently fixed, and immersed them in AF-mixed SoROCS. Fluorescent microspheres, located at a depth ~1 mm from the surface of the noodles, excited by 940-nm laser and fluorescently labelled gluten with AF excited at 840-nm, were then imaged (Fig. 3m, n). The experimental data on the fluorescence intensity I of fluorescent microspheres (point spread function) were well represented by

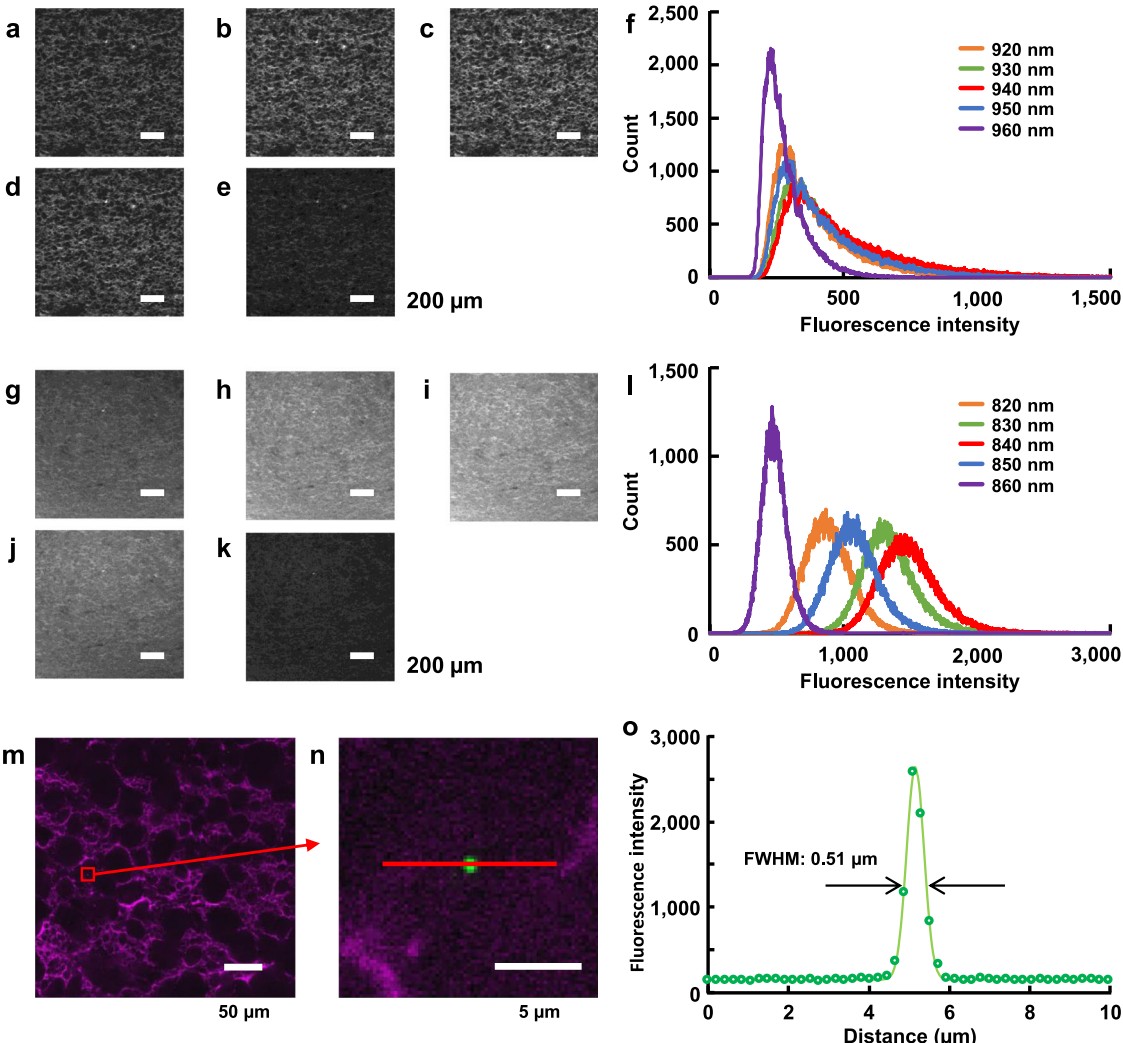

**Fig. 3 Two-photon excitation microscopy imaging of gluten.** 2D gluten structure of noodle treated with SoROCS mixed with Thiolite™ Green (**a–e**) or Alexa Fluor™ 633 C$_5$ Maleimide (**g–k**) after fixation. Note that the image processing was performed in the same manner for each set (**a–e**, **g–k**) to improve visibility. Scale bars indicate 200 μm (**a–e**, **g–k**). 2D images were excited at 920 nm (**a**), 930 nm (**b**), 940 nm (**c**), 950 nm (**d**), 960 nm (**e**), 820 nm (**g**), 830 nm (**h**), 840 nm (**i**), 850 nm (**j**) or 860 nm (**k**). **f**, **l** Fluorescence intensity histograms were created using a 12-bit image obtained from a two-photon excitation microscopy (2PEM). The histograms are shown in (**f**) for 920 nm excitation (orange), 930 nm (green), 940 nm (red), 950 nm (blue) and 960 nm (purple), respectively, and in (**l**) for 820 nm (orange), 830 nm (green), 840 nm (red), 850 nm (blue) and 860 nm (purple), respectively. Fluorescence intensities of ~1500 or ~3000 are shown for (**f** or **l**), respectively. **a–e**, **g–k** This experiment was repeated three times using independently prepared noodle samples, with results similar to those shown in (**f**, **l**), respectively. **m** Merged image of gluten (magenta) and fluorescent microspheres (green) obtained using 2PEM. **n** Enlarged image of the area marked with a red square in (**m**). Scale bar in (**m**, **n**) indicates 50 and 5 μm, respectively. **o** Plot profile of fluorescence intensity (point spread function) for fluorescent microspheres along a red line shown in (**n**). Plots and line shown in (**o**) indicate experimental data on the fluorescence intensity and calculated data using Gaussian function presented as Eq. (2) ($I_0 = 156.2$, $x_c = 5.16$ μm, $A = 1346.9$ and $w = 0.509$ μm), respectively. Two black opposing arrows shown in (**o**) indicate FWHM. **m–o** This experiment was repeated three times using independently prepared noodle samples (the mean value and SD for FWHM was 0.54 and 0.03 μm, respectively). Source data underlying (**f**, **l**, **o**) are provided as a Source Data file.

the Gaussian function presented as Eq. (2) (Fig. 3o). Since the obtained value of the full width at half maximum (FWHM) of the peak was similar to the diameter of the fluorescent microspheres, we determined that SoROCS-treated samples can be imaged with the resolution of 0.54 ± 0.03 μm (the value shows mean ± SD for FWHM obtained from three independently prepared noodle samples).

Having established clearing and imaging conditions, we performed whole-noodle imaging with TG-mixed SoROCS (Fig. 4). With the 2PEM, an imaging depth of over 2 mm can be achieved (Fig. 4b, c), although SoROCS caused a ~2.23-fold linear expansion (Fig. 2a). In contrast, the existing depth imaging that does not cause the sample to become transparent is limited to depths up to ~50 μm (Fig. 4b, c).

Moreover, clearing with SoROCS made it possible to observe the entire structure of gluten formed around starch granules with a size of ~30 μm. Furthermore, to examine the fluorescent image quality of gluten depending on the depth from the surface of the noodles, the variance VAR was calculated from the histogram according to Eq. (3), which characterises the distribution of fluorescence intensity by level and indicates a smaller value when the fluorescence intensity is concentrated on a specific value. The VAR value became smaller with increasing depth from the surface of the noodles (Fig. 4d); at a depth of 0.550 mm, the VAR value was $5.69 ± 0.31 × 10^5$, and when the depth exceeded 2 mm, the VAR value decreased significantly to $8.98 ± 3.81 × 10^4$. However, the skeletal structure of gluten can be confirmed even at a depth of over 2 mm (Fig. 4d–h).

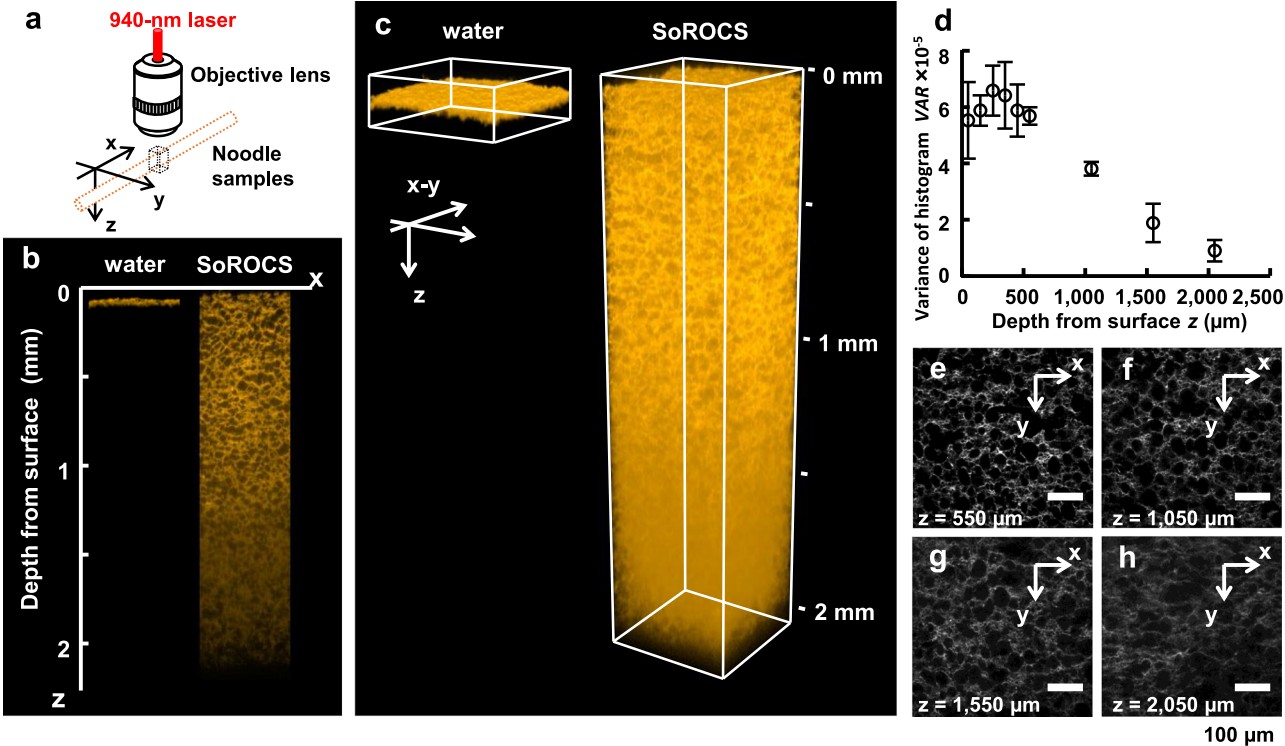

**Fig. 4 3D reconstruction of gluten in whole noodle. a** Experimental setup for 2PEM imaging. **b** Comparison of deep imaging of noodles treated with water (left) and SoROCS (right). **c** 3D reconstruction of gluten for noodle treated with water (left) and SoROCS (right). Note that when rendering with FluoRender, gamma change (3.2) and alpha blending (500) were applied and a threshold was set to 146 and 90 for water and SoROCS, respectively (**b**, **c**). **d** The fluorescent image quality of gluten depending on the depth from the surface of the noodles. Data are presented as mean values ± SD ($N = 3$ independently prepared noodle samples). One of the section images obtained from these three different samples is shown in (**e–h**), respectively. Note that the 2D imaging conditions with 2PEM were adjusted to be optimal for each depth (**d–h**). Optical sections at several depths: $z = 550\ \mu m$ (**e**), $z = 1050\ \mu m$ (**f**), $z = 1550\ \mu m$ (**g**) and $z = 2050\ \mu m$ (**h**). Scale bars indicate 100 μm (**e–h**). Source data underlying (**d**) are provided as a Source Data file.

**Dramatic changes in gluten structure.** Our successful imaging of a whole noodle using SoROCS led us to demonstrate the imaging of other gluten skeletal structures. Then, we prepared the noodles by adding gluten powder or sodium chloride to wheat flour. Adding gluten powder to flour at 10% or 20% weight ratio aimed to create an additional auxiliary structure between the gluten skeleton formed in the wheat dough, where the noodle thus produced is denoted as 10%- or 20%-gluten noodle, respectively. As expected, compared to the noodles without any addition, which are denoted as control noodles, the addition of gluten powder did not change the main skeletal structure composed of gluten, but the overall structure became hazy (Fig. 5a–c). To illustrate this, we binarised the optical section image to calculate the percentage of gluten in the image, which was found to be 13.27% ± 1.01%, 21.69% ± 1.02% and 29.60% ± 0.68% (the value shows mean ± SD for $N = 3$ independently prepared noodle samples) for control noodle, 10%-gluten noodle and 20%-gluten noodle, respectively, indicating that increase amounts of gluten powder caused an increase in the formation of gluten networks (Fig. 5d–i). In contrast, the addition of sodium chloride at weight ratios of 3%, 6%, 9% and 12% was intended to alter the skeletal structures of gluten, and the noodles thus produced are denoted as 3%-NaCl, 6%-NaCl, 9%-NaCl and 12%-NaCl noodles, respectively. While there are several possible explanations for the mechanism by which sodium chloride alters the skeletal structure of gluten, sodium chloride likely shields charges on the gluten, thus limiting electrostatic repulsion between gluten polymers and causing them to aggregate[27]. As the sodium chloride concentration increased to 3 and 6%, the structure of gluten dramatically changed from a relatively uniform honeycomb-shaped network to

a sparse large clump (Fig. 5j, k). Subsequently, when the sodium chloride concentration was further increased to 9 and 12%, the aggregated large clumps were dispersed, although these concentrations were not realistic (Fig. 5l, m).

Next, to examine the effect of the changes in the skeletal structures on the mechanical properties, we performed a compression test (Fig. 5n). For 10%- and 20%-gluten noodles, the stress-strain curve obtained when compressing the noodles with a wedge-shaped plunger shifted to the upper side. Conversely, the stress-strain curves of the noodles containing sodium chloride moved downward. These results suggest that reinforcement of the gluten skeletal structure increases structural strength, but cleavage of the gluten network decreases structural strength. Notably, the dramatic change in the main skeletal structure of gluten had a great impact on the mechanical properties of wheat dough. These results are consistent with previous research on the physicochemical and structural changes of gluten[28]. The high-resolution 3D reconstruction of SoROCS may facilitate structural insight into the mechanical properties of wheat dough.

As a further demonstration of this imaging methodology with SoROCS, we quantified gluten volume employing 3D measurement. First, the fluorescence intensity was binarised with respect to the 3D image of gluten in 12%-NaCl noodles (Fig. 6a, b). Each gluten clump that is three dimensionally isolated from the others is represented by a different colour in Fig. 6b. Notably, we can readily observe the positional relationship of each gluten clump from any angle and direction (Fig. 6c). Next, when the volume distribution of gluten was analysed, a small number of huge gluten clumps were found to occupy most of the total product (Fig. 6d). Information on the volume distribution that can only be

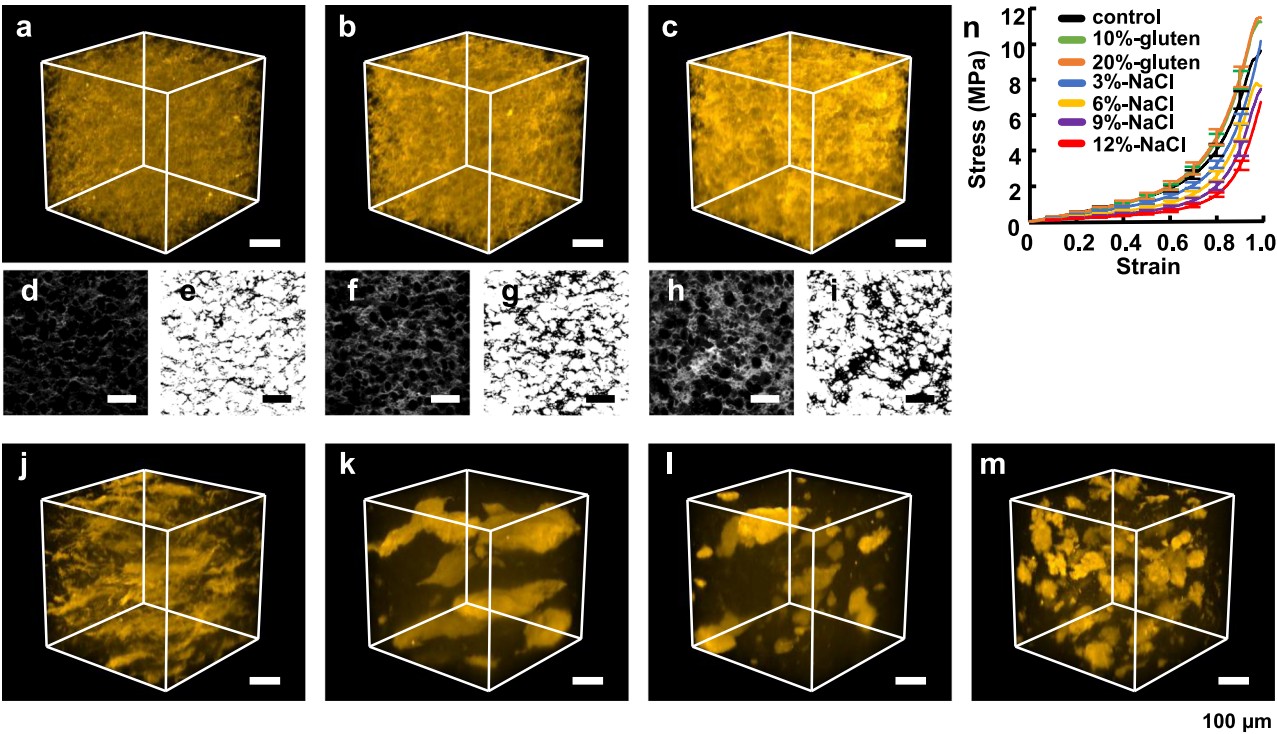

**Fig. 5 3D structure of gluten and stress-strain curves for various noodles. a–c, j–m** Skeletal structures of gluten labelled with Thiolite™ Green were imaged with 2PEM. The 3D structure of gluten in control noodle (**a**), 10%-gluten noodle (**b**), 20%-gluten (**c**), 3%-NaCl (**j**), 6%-NaCl (**k**), 9%-NaCl (**l**) and 12%-NaCl (**m**). **d, f, h** Optical sections images in (**a–c**) and binarised images obtained by the Otsu method (**e, g, i**). The percentage of gluten calculated by Otsu method for control noodle, 10%-gluten noodle and 20%-gluten noodle was 13.27 ± 1.01, 21.69 ± 1.02 and 29.60 ± 0.68, respectively (the values show mean ± SD for $N = 3$ independently prepared noodle samples). Scale bars indicate 100 μm (**a–m**). Note that when rendering with FluoRender, alpha blending (500) was applied (**a–c, j–m**). The stress-strain curves are shown in (**n**) for control (black), 10%-gluten (green), 20%-gluten (orange), 3%-NaCl (blue), 6%-NaCl (yellow), 9%-NaCl (purple) and 12%-NaCl (red). The strain was normalised by the thickness of each sample, and the stress value for every 0.1 strain from 0.1 to 0.9 was calculated by the interpolation complement method. Data are presented as mean values ± SD ($N = 20$ different samples taken from the same noodle preparation) (**n**). Source data underlying (**n**) are provided as a Source Data file.

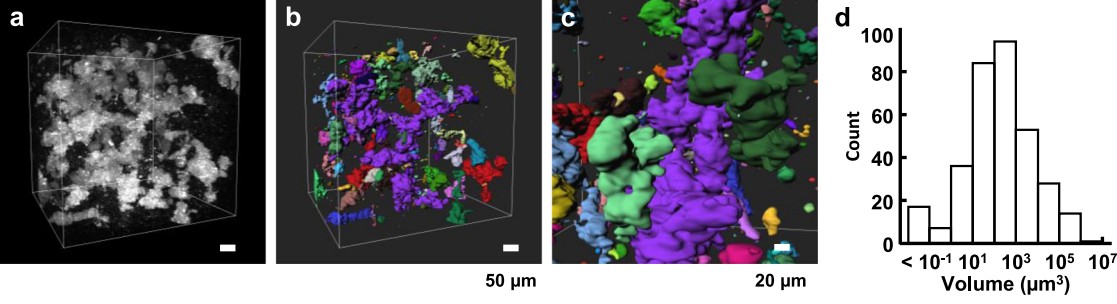

**Fig. 6 Volumetric analysis of a 3D image. a** 3D rendering of gluten in a 12%-NaCl noodle. Note, the 3D data set shown in this figure is the same as that in Fig. 5m. **b** Surface rendering by binarising the fluorescence intensity of (**a**). Each gluten clump that is three dimensionally isolated from others is displayed in a different colour in (**b**). **c** Image rendered from different viewpoints of the 3D image shown in (**b**). Scale bars indicate 50 μm (**a, b**) and 20 μm (**c**). **d** Volume distribution of gluten shown in (**b**). Source data underlying (**d**) are provided as a Source Data file.

obtained by this imaging methodology with SoROCS will serve to improve the industrial performance of wheat flour foods.

**Imaging using confocal laser scanning microscopy.** 2PEM is a powerful tool for deep imaging, but it is not accessible to all researchers due to its high equipment cost. Thus, we demonstrated confocal laser scanning microscopy (CLSM) observation of SoROCS-treated noodles to evaluate imaging performance. With CLSM, we obtained cleared noodle images (Fig. 7a, b). The xy image at a depth of 250 μm from the noodle surface depicts clear gluten structures (Fig. 7c). To examine the fluorescent image quality of gluten, the VAR was calculated from the histogram for

xy images at several depths (Fig. 7c–e). The VAR value tended to decrease as the depth from the surface of the noodles increased, although there was no significant difference observed (Fig. 7e). The working distance (WD) of the objective lens used in this study limits the imaging depth to 600 μm, however, given that the gluten that surrounds starch granules of ~30 μm in size forms a honeycomb-shaped network, SoROCS is compatible with imaging using CLSM as well as 2PEM.

## Discussion

We have developed an optical clearing reagent, SoROCS, which makes wheat-based products transparent in a few days and

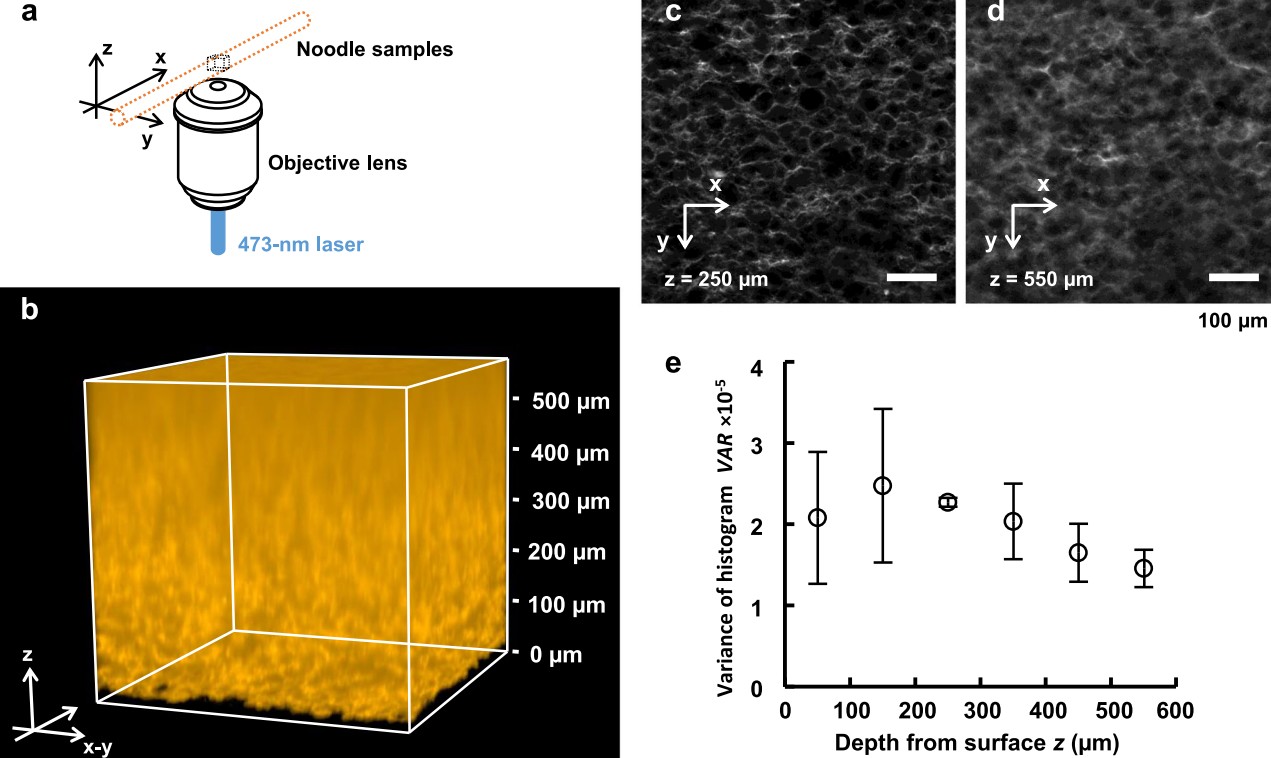

**Fig. 7 3D reconstruction of gluten in a noodle observed by confocal laser scanning microscopy. a** Experimental setup for confocal laser scanning microscopy (CLSM) imaging. **b** 3D reconstruction of the gluten labelled with Thiolite™ Green. Note that when rendering with FluoRender, gamma change (3.2), alpha blending (500) and threshold (90) were applied (**b**). Optical sections at several depths: $z = 250$ µm (**c**) and $z = 550$ µm (**d**). Scale bars indicate 100 µm (**c**, **d**). **e** The fluorescent image quality of gluten depending on the depth from the surface of the noodles. Data are presented as mean values ± SD ($N = 3$ independently prepared noodle samples). Note that the 2D imaging conditions with CLSM were adjusted to be optimal for each depth (**c–e**). One of the images used to calculate the VAR values in (**e**) is shown in (**c**, **d**). Source data underlying (**e**) are provided as a Source Data file.

enables imaging of the gluten 3D structure at the size on a millimetre scale and at submicron resolution. The main advantage of SoROCS is the ease with which optical clearing can be achieved only by immersing. Moreover, the imaging devices for the SoROCS-treated sample can be applied not only to the 2PEM but also to a CLSM, which many researchers can easily access. In addition, optical sectioning allows us to view the images of desired regions and planes in any orientation, which helps us to better understand the gluten structure of samples. Meanwhile, similar to the previously reported incipient optical clearing reagents for biological samples[16], the significant changes in sample volume caused by clearing could be an artefact of imaging, although the expansion of the samples did not affect the skeletal structure of gluten. Nevertheless, suppression of sample expansion should be addressed in the next step of SoROCS development. However, even the current SoROCS will contribute to bridge the imaging gap between the molecular level phenomenon of gluten and the edible quality of wheat-based products. Our findings highlight the potential of SoROCS as an alternative to the currently widespread imaging methods for viewing the gluten structure in wheat dough and other cereal-based products. We anticipate that our imaging technique will be a starting point for identifying and modelling the microstructural characteristics of gluten towards improving the industrial performance of wheat flours.

## Methods

**Preparation of SoROCS.** To prepare the SoROCS, sodium salicylate (Nacalai tesque, 31821-45) was dissolved completely in Milli-Q water under stirring and heating at 70 °C. Then, Triton X-100 (Nacalai tesque, 35501-02) was added to 4 M

sodium salicylate solution to a concentration of 0.1% (wt wt$^{-1}$). RI were determined at 30 °C using a refractometer (Atago, PAL-BX/RI). For fluorescent labelling of gluten, TG (AAT Bioquest, 21508) and AF 633 $C_5$ Maleimide (Invitrogen, A20342) dissolved in dimethyl sulfoxide (Wako Chemicals, 043-07216) at a concentration of 10 mM was added to the SoROCS to yield a final concentration of 0.0005% and 0.0002% (wt wt$^{-1}$), respectively.

**Noodle preparation.** Noodles with no excess gluten powder or sodium chloride, which were denoted as control noodles, were prepared by mixing 500 g of wheat flour for noodle making (Nisshin Seifun, crude protein 8.5%, ash 0.34%) with 160 g of Milli-Q water, followed by kneading using a mixer (Hobart, N50) for 20 min at 139 rpm to produce a wheat dough (Supplementary Fig. 4a). The wheat dough was then put into a pasta-making machine (Bottene) equipped with a vacuum pump (Ulvac, DTC21) and extruded through an opening in a Teflon die with a width of 1.8 mm and a thickness of 0.7 mm (tagliolini No. 17) at 11.0–15.0 kPa (abs) to produce noodles (Supplementary Fig. 4b). The compositions of noodles containing excess gluten powder (Nacalai tesque, 17011-95) or sodium chloride (Nacalai tesque, 31320-05) are listed in Supplementary Table 2. The gluten powder was mixed with the wheat flour at 10% and 20% weight ratio before adding Milli-Q water, where the noodle thus produced was denoted as 10% and 20% gluten, respectively. Sodium chloride was dissolved in Milli-Q water at weight ratios of 3, 6, 9 and 12% before mixing with the wheat flour, where the noodle thus produced was denoted as 3%-NaCl, 6%-NaCl, 9%-NaCl and 12%-NaCl, respectively. The amount of Milli-Q water was changed depending on the amount of sodium chloride.

**Fixation and optical clearing.** The noodle samples were fixed in 4% (wt wt$^{-1}$) paraformaldehyde in PBS for 1.5 h at 30 °C, washed twice in D-PBS and immersed in ClearSee, SoROCS or water (0.5% (wt wt$^{-1}$) sodium azide added to prevent spoilage) at 30 °C with shaking for 3 days. ClearSee was prepared by mixing xylitol powder (Nacalai tesque, 36718-65; final 10% (wt vol$^{-1}$)), sodium deoxycholate (Nacalai tesque, 10712-96; final 15% (wt vol$^{-1}$)) and urea (Nacalai tesque, 35905-35; final 25% (wt vol$^{-1}$)) in Milli-Q water[19].

**Measurement of opacity.** The opacity of the noodles was determined using a cooled CCD camera system (GE Healthcare, ImageQuant LAS 4000). The sample

was placed on a glass dish with the mid-plane on the bottom filled with ClearSee, SoROCS or water (0.5% (wt wt$^{-1}$) sodium azide added to prevent spoilage) as a reference. Then, the sample was digitised by the CCD camera, which was cooled to −25 °C with a lens (F0.85 43 mm), using white transillumination. The grey value of the 3072 × 2048-pixel 16-bit tiff image was obtained with an exposure time of 6 s and an iris of F2.8. The opacity of the sample was calculated by normalising the grey value of the sample with that of the background using free software (Fiji[29]). The contrast C, which represents the difference in grey level, was calculated according to the following equation from an image of noodles laid with a film on which a grid pattern was printed:

$$C = \frac{g_{max} - g_{min}}{g_{max} + g_{min}} \tag{1}$$

where $g_{max}$ and $g_{min}$ represent the maximum and minimum values of the grey level, respectively, in a 5-mm long plot profile. The plot profile was measured using Fiji[29] software.

**Measurement of sample morphology.** For the measurement of sample linear expansion, photographs of 5472 × 3648 pixels were taken of the top of the samples placed on a glass dish with the mid-plane on the bottom filled with each clearing reagent using a high-resolution digital camera (Canon, EOS 70D; ISO = 100, aperture value = 29, shutter speed = 0.5) with a lens (Canon, EFS 60 mm). Then, the aberration of image-forming capability (peripheral illumination, colour blur and distortions) and the deterioration of the resolution resulting from diffraction phenomena in the photographs were corrected using software (Canon, Digital Photo Professional) based on the designed-lens value provided by the supplier. The 14-bit raw data were recorded in the 16-bit tiff format. Based on the images of sample width, linear expansion was determined using the software (Fiji[29]).

**Two-photon excitation microscopy imaging.** Samples were imaged using an upright or inverted 2PEM system (Olympus, upright: FV1200MPE-BX61WI, inverted: FV1000MPE-IX83) with a ×25 immersion objective lens for SoROCS (XLSLPLN25XGMP, a numerical aperture (NA) = 1.00, WD = 8 mm, RI = 1.41–1.52, a correction collar), a ×25 immersion objective lens for water (XLPlan N 25X, NA = 1.05, WD = 2 mm, a correction collar) or a ×10 dry objective lens (UPLSAPO10X2, NA = 0.40, WD = 3.1 mm). The brightness compensation function in the z direction was used to change detector sensitivity and laser power. Fluorescence signals were quantified with a normal PMT for AF or GaAsP PMT for TG. Imaging was carried out with Fluoview FV10-ASW software Ver. 4.2c (Olympus). TG and AF were excited at 940 and 840 nm, respectively, using InSight DeepSee (Spectra-Physics). Objective lenses used in this study, as well as the imaging conditions for all figures, are summarised in Supplementary Table 3.

**Point spread function analysis.** Fluorescent microspheres (yellow green fluorescent FluoSpheres, diameter = 0.49 μm, Invitrogen, F8813) were diluted in Milli-Q water at a concentration of 0.01% (vol vol$^{-1}$) and dispersed under ultrasonic waves for 15 min. Next, 160 g of water containing fluorescent microspheres was mixed with 500 g of wheat flour to prepare noodles in the same manner as described above. The noodles were fixed in 4% (wt wt$^{-1}$) PFA/PBS for 1.5 h at 30 °C and immersed in AF-mixed SoROCS at 30 °C with shaking for 3 days. Subsequently, 2PEM xy images of AF-labelled gluten and fluorescent microspheres at ~1 mm depth from the noodle surface were acquired at 840- and 940-nm excitation, respectively, as described above. The experimental data on the fluorescence intensity I of fluorescent microspheres (point spread function) was represented by the following Gaussian function:

$$I = I_0 + \frac{A \times \exp\left\{-4\ln(2)(x - x_c)^2 / w^2\right\}}{w\sqrt{\pi / 4\ln(2)}} \tag{2}$$

where $I_0$ is the baseline fluorescence intensity, $x_c$ is the centre position of the peak, A is the coefficient and w is the FWHM of the peak. Gaussian fitting was used to obtain the FWHM values for x axes using the OriginPro 8.1J SR3 software (OriginLab, Ver. 8.1.34.90).

**Fluorescent image quality analysis.** To examine the fluorescent image quality of gluten depending on the depth from the surface of the noodles, the variance VAR was calculated from the histogram according to the following equation:

$$VAR = \sum \left(I - I_p\right)^2 F(I) \tag{3}$$

where I is the fluorescence intensity, $I_p$ is the fluorescence intensity at the peak top of the histogram and F(I) represents the normalised frequency for each fluorescence intensity I. Note that the 2D imaging conditions with 2PEM and CLSM were adjusted to be optimal for each depth, and the values of fluorescence intensity in blocked up shadows and blown out highlights were excluded when calculating the VAR values.

**Confocal laser scanning microscopy imaging.** The samples were imaged using an inverted confocal laser scanning microscopy (CLSM) system (Olympus, FV1000D IX81) with a ×20 dry objective lens (UPLSAPO 20X, NA = 0.75, WD = 0.6). The

brightness compensation function in the z direction was used to change detector sensitivity and laser power. Imaging was carried out with Fluoview FV10-ASW software Ver. 3.0a (Olympus). TG and AF 594 were excited at 473 and 559 nm, respectively, using diode lasers. The imaging conditions for all figures are summarised in Supplementary Table 3.

**Protein network analysis.** Software (National Cancer Institute, National Institute of Health, AngioTool64 Ver. 0.6a) was used to analyse the protein network characteristics for CLSM and 2PEM images of gluten skeletal structures. The same settings and parameters were applied to ensure reproducible quantification of the gluten network: Vessel thickness was set to 8, intensity low and high threshold was set to 0 and 255, respectively, small particles were removed under 30 pixels and the function of fill holes was deactivated.

**Immunohistochemistry staining.** The noodle samples were fixed in 4% (wt wt$^{-1}$) PFA/PBS for 1.5 h at 30 °C and washed twice in D-PBS. The samples were then embedded in OCT compound, sectioned at a thickness of 10 μm at −20 °C using a cryostat (Leica, CM1860) and collected on slides. The OCT compound on the slide was then rinsed with distilled water. Subsequent immunohistochemistry staining was performed at room temperature. The slides were washed twice in D-PBS, blocked for 30 min using SuperBlock™ (PBS) blocking buffer (1:4 (D-PBS); Pierce Thermo scientific, 37515) and incubated for 1 h with primary antibodies (1:1000 (D-PBS); anti-wheat gluten chicken-polyclonal IgY, Agrisera AB, AS09-571). The slides were then washed twice with D-PBS and incubated for 1 h with a secondary antibody (1:200 (D-PBS); goat anti-chicken IgY with Alexa Fluor 594, Invitrogen, A11042). Next, the slides were washed twice in D-PBS and incubated for 30 min with TG (10 mM in DMSO) diluted with D-PBS at a final concentration of 0.05% (wt wt$^{-1}$). Fluorescent signals for TG and Alexa Fluor 594 were acquired at 473- and 559-nm excitation, respectively, using the CLSM. The imaging conditions for all figures are summarised in Supplementary Table 3.

**Image reconstruction and re-slicing.** Software (FluoRender) was used to visualise the 3D reconstructed image and obtain various 2D stacked images from an entire 3D data set for Figs. 4b, c, 5a–c, j–m and 7b. Imaris® x64 software (Bitplane, Ver. 8.3.1) was used to render the 3D data set and to analyse the volume distribution presented in Fig. 6.

**Compression test.** Mechanical property was characterised using a stress-strain curve, which showed the relationship between stress and strain during compression of the noodle samples. The stress-strain curves were generated using a creep metre (Yamaden, RE2-33005B Rheoner II) equipped with a wedge-shaped plunger (No. 49) and a 20 N load cell at 0.1 mm s$^{-1}$. A noodle sample cut to ~30 mm in length was placed on a cylindrical stand. Then, the force applied by the wedge-shaped plunger and the distance travelled were recorded in the computer while the cylindrical stand rose. The strain normalised by the thickness of the sample was set to 99%.

**Statistical analysis.** Calculation of numerical data and statistical analysis was performed using Microsoft 365 Excel. Data are presented as mean ± SD or mean ± SEM as indicated. P values were calculated using the Student's two-tailed unpaired t test. A P < 0.05 was considered statistically significant.

**Reporting summary.** Further information on research design is available in the Nature Research Reporting Summary linked to this article.

## Data availability

Data supporting the findings of this work are available within the paper and its Supplementary Information files. The materials generated and analysed during the current study are available from the corresponding author upon request. The image data sets are available at Zenodo (https://zenodo.org/record/4540950#.YDAuLlVKipo)[30]. Source data are provided with this paper.

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

## Acknowledgements

We acknowledge Fumito Tani (Kyoto University) for experimental comments. We also acknowledge Michiyuki Matsuda (Kyoto University) for advice and technical support. We thank Kanako Takakura (Kyoto University) for providing technical assistance with 2PEM observation. We also thank Kousaku Ohinata and Yukako Hayashi (Kyoto University) for advice and technical support with the immunohistochemistry staining. This work was supported by JSPS KAKENHI Grant Numbers JP14J02443, JP18K14421, JP16H06280 (ABiS), Toyo Institute of Food Technology Foundation, Sapporo Bioscience Foundation, The Tojuro Iijima Foundation for Food Science and Technology and The Foundation for Dietary Scientific Research. Microscopy analysis using 2PEM was supported by Kyoto University Live Imaging Center. This work was made possible in part by software (FluoRender) funded by the NIH (FluoRender: Visualisation-Based and Interactive Analysis for Multichannel Microscopy Data, 1R01EB023947-01) and the National Institute of General Medical Sciences of the National Institutes of Health under grant number P41 GM103545-18.

## Author contributions

T.O. and Y.M. conceived the study, T.O. performed the experiments, analysed data and wrote the paper and Y.M. discussed the data with T.O. and edited the paper.

## Competing interests

The authors declare no competing interests.
