## [Peer Review File · Nature Communications]

REVIEWER COMMENTS

Reviewer #1 (Remarks to the Author):

This manuscript reports about a methodology to improve the 3D imaging of the structure of gluten. The researchers have identified an optical clearing agent (SoROCS) which contributes to an increase in transparency. The main conclusions can be resumed as follows:

- The identified optical clearing agent makes wheat-based products transparent in a few days;
- Applying this agent, the 3D structure of gluten could be visualized in the mm scale at submicron resolution.
- The demonstrated visualization techniques are two-photon spectroscopy and confocal laser scanning microscopy;
- Changes in the network structure of the gluten (e.g. due to stress addition) could be monitored.

The paper contains important information for an improved quality monitoring of wheat-based products. As such I consider this manuscript worth publishing. However, before the paper can be considered for publication in this journal, some parts need revision. The topics that need to be addressed in any revised version are as follows:

- One of the sentences written in the abstract is confusing. It is mentioned that there is 'a lack of imaging techniques performed in situ under non-destructive conditions'. The words 'in situ' and 'non-destructive' are confusing since at the end the here proposed technique is in my opinion destructive (since you apply a dye) and is sample-based. When I read the word 'in situ' in the abstract I got the impression that a kind of in-line monitoring system would be proposed.
- The last sentence in the introduction 'Therefore, adjusting the RI ... due to the low water absorption of starch' needs some clarification. I currently don't understand the link with the previous sentences.
- I would add a reference to the sentence 'A high pH will denature proteins'.
- It was not clear to me how the reported concentrations of Triton X-100 and sodium salicylate were selected.
- Isn't there a risk that the swelling of the starch granules during the gelatinisation process imparts the gluten structure that you envision to study?
- You mention that you quantified the opacity of the 'untreated' and 'treated' noodles based on the transmitted light intensity. This is a correct approach but could you please also add the numbers to the text. A similar remark is valid for the reporting of the linear expansion values.
- The evaluation of the fluorescence image quality is described in a qualitative way. Why not quantifying the contrast values by taking a cross section and calculating the contrast?
- The 2 previous remarks can be generalized. All the results need to be described in the text in a more quantitative way. This includes the depth of imaging (cfr. 'a longer excitation wavelength enables deeper imaging') and the applied laser powers.
- In the conclusion it is mentioned that the findings of this paper can be seen as an alternative for the currently widespread imaging methods for viewing the gluten structure in wheat dough. Can you benchmark your results with the state-of-the-art results? Also with respect to the two researched spectroscopic systems I think it is worth to add a conclusion comparing the performances of both methods for the gluten structure visualization.
- It is reported that the measurements were repeated 20 times. Please elaborate if these were 20 independently prepared samples, if these were 20 samples taken from the same preparation step or if

these are 20 measurements made on the same sample. The same holds for n=3 mentioned in the caption of figure1.

- It is hard to visually observe a difference for the images shown in the extended data figure 2.
- Extended data figure 3 shows a high similarity with Figure 3. What is the added value?

Reviewer #2 (Remarks to the Author):

Ogawa and Matsuura developed an optical clearing reagent (SoROCS) to render whole-noodle imaging of wheat. SoROCS enabled 3D imaging of gluten structure in the intact noodles. The manuscript is well written and clearly describes the experimental procedure. But I could not understand the novelty and importance of this method, while my field of expertise is plant cell biology and imaging.

The authors should introduce previous research on gluten of the field and properly demonstrate the superiority of the methods they have developed by showing the data.

What methods have been used to analyze gluten structure? Electron microscopy?

How is it different from that?

Are there any specific novelties that the newly developed methods will reveal? The authors should also provide data to show its novelties.

I have outlined the major concerns in detail below:

Major concerns:

What does "intact" mean? The authors only show the data after swelling, but is there any evidence demonstrated that it is "intact"? What rationale do the authors have that sodium salicylate treatment keeps the noodle intact without fixation?

P4, L16-17; What kind of serendipity was it?

It would be helpful for the reader to have a specific explanation of why the authors tried sodium salicylate.

Fig. 1 or P6, L7-9; The superiority of SoROCS cannot be determined without comparison with other transparency reagents.

P7, L13-17; The authors claim to have seen the structure of gluten, but is gluten specifically stained? Is it possible to confirm the specificity of visualized the gluten by other methods?

Since the sample is swollen, the authors should also indicate the original scale.

"Expansion Microscopy (Chen et al. 2015, Science)" is a good example of how to show the scale for both original and swollen.

P8, L17-P9, L5; I didn't understand the need for deep or whole imaging. The authors said that "when comparing the xy images at several depths, they show almost the same structure". If so, whole-imaging is no need?

P8 L4-L5, P9 L5, P9 L18-P10 1; "TG and AF may render more sharply", almost same structure", and "the overall structure became hazy" are "too" subjective. The authors should provide quantitative data from these images.

P9 L7-8; the authors claimed that "this is the first report that renders the 3D structure of gluten, especially in millimetre-scale sizes and submicron resolution.", but which data indicates submicron resolution?

P19 L18 and Fig 4; The authors should use for the same objective lens or same magnification to compare between two-photon and confocal microscopy. Why did the authors choose a 10x objective lens?

Fig 4; I could not see the fluorescent signals of the samples near the glass surface, is that a bumpy noodle?

It doesn't look that bumpy in the other figures.

Minor concerns:

P16 L4, How many degrees for heating to melt?

P16 L7-L10; please provide the stock concentration of each dye in DMSO.

P9 L1-L2 and P17 L5; The authors said that "SoROCS caused an approximately 2.75-fold linear expansion" and the thickness of noodles was 0.78 mm, but Fig2c showed the thickness was over 2.5 mm. What is this difference?

Extended Data Fig.2a-e, g-k; It's too dark to observe the staining.

Reviewer #3 (Remarks to the Author):

The manuscript by Ogawa and Matsumura reports on a new methodology to visualise gluten network in the presence of starch. To allow this, they have established a clearing reagent and new imaging conditions. As a result, the manuscript is more akin to a new experimental protocol. The manuscript is well written and presents new information and has, in my opinion, good chances to be accepted by the relevant research community, as the technique is relatively simple and standard equipment (e.g., CLSM) may be used to carry out the visualisation. In addition, it may help other researchers to establish structure-function relationships in samples that are otherwise difficult to visualise (e.g., bread dough or laminated dough formulations).

I have a few points that if they are addressed, in my opinion, may help to improve the presentation of the work:

1) The length scale that the structure is observed is in the range of $\sim 100 \mu\text{m}$. This needs to be clearly mentioned in the manuscript and contrasted with other techniques that provide "deeper" visualisation of the structure. For example, what is the structure of EACH strand that is observed in Figure 2, 3 and 4? Unfortunately, it is not possible to obtain this information with this technique.

2) The present visualisation method has not been compared with other previously published work. How does the new technique compare with already published work? Has new structural information been revealed or it is just an additional experimental technique in the toolkit of the researchers?

3) There is a section with advantages of SoROCS. A section with disadvantages and limitations also needs to be present, as there is no such technique without its disadvantages.

Minor comments

Lines 112-113: Gelatinisation cannot be achieved without heating. Do you mean "swelling"? Please, rephrase.

Lines 181-185: This information is well documented and routinely observed in the literature. Please, cite key past work to complete the discussion of this part.

Please, explain the purpose of Triton in your clearing reagent.

Figure 3. What is the size of the scalebar?

Title needs to be revised as it is not informative. A "giant protein" should be replaced by "gluten".

Dr Vassilis Kontogiorgos

Dear Reviewer #1

We sincerely thank you for your valuable comments regarding our manuscript submitted to *Nature Communications* (NCOMMS-20-22137-T). We also appreciate the time and effort you have dedicated to providing insightful feedback on ways to strengthen our paper. Revisions have been made according to your comments, which are indicated in **red text throughout the revised manuscript**.

Responses to the Comments (in *blue italics* the reviewer's comment):

This manuscript reports about a methodology to improve the 3D imaging of the structure of gluten. The researchers have identified an optical clearing agent (SoROCS) which contributes to an increase in transparency. The main conclusions can be resumed as follows:

- The identified optical clearing agent makes wheat-based products transparent in a few days;*
 - Applying this agent, the 3D structure of gluten could be visualized in the mm scale at submicron resolution.*
 - The demonstrated visualization techniques are two-photon spectroscopy and confocal laser scanning microscopy;*
 - Changes in the network structure of the gluten (e.g. due to stress addition) could be monitored.*
- The paper contains important information for an improved quality monitoring of wheat-based products. As such I consider this manuscript worth publishing. However, before the paper can be considered for publication in this journal, some parts need revision.*

RESPONSE: We would like to express our sincere gratitude for your evaluation of our paper. We have incorporated changes that reflect the detailed suggestions you have graciously provided. We also hope that our edits and the responses we provide below satisfactorily address all of the issues and concerns you have noted.

The topics that need to be addressed in any revised version are as follows:

- 1. One of the sentences written in the abstract is confusing. It is mentioned that there is 'a lack of imaging techniques performed in situ under non-destructive conditions'. The words 'in situ' and 'non-destructive' are confusing since at the end the here proposed technique is in my opinion destructive (since you apply a dye) and is sample-based. When I read the word 'in situ' in the abstract I got the impression that a kind of in-line monitoring system would be proposed.*

RESPONSE: In response to your comment, the corresponding sentence "performed in situ under non-destructive conditions" has been revised to "for complex systems

composed of giant macromolecules” (P. 1, L. 16).

2. *The last sentence in the introduction ‘Therefore, adjusting the RI ... due to the low water absorption of starch’ needs some clarification. I currently don’t understand the link with the previous sentences.*

RESPONSE: As you mentioned, the link with the previous sentences was ambiguous. We have, therefore, deleted the second half of the sentence. Instead, we have conducted a new comparative test with the previous clearing reagents (P. 7-8, L. 109-137).

3. *I would add a reference to the sentence ‘A high pH will denature proteins’*

RESPONSE: According to your comment, the reference has been added with minor revisions to the text (P. 6, L. 93, P. 21, L. 361-362).

4. *It was not clear to me how the reported concentrations of Triton X-100 and sodium salicylate were selected.*

RESPONSE: In response to your comment, a description regarding how the concentration of Triton X-100 and sodium salicylate were selected has been added (P. 6, L. 97-98).

5. *Isn’t there a risk that the swelling of the starch granules during the gelatinisation process imparts the gluten structure that you envision to study?*

RESPONSE: As you correctly point out, the swelling of the starch granules may alter the gluten structure. Therefore, we have conducted a new study on the effect of swelling of the starch granules on the gluten structural characteristics. Protein network analysis revealed that swelling does not alter the structural or morphological network attributes of gluten (P. 9-10, L. 151-179).

6. *You mention that you quantified the opacity of the ‘untreated’ and ‘treated’ noodles based on the transmitted light intensity. This is a correct approach but could you please also add the numbers to the text. A similar remark is valid for the reporting of the linear expansion values.*

RESPONSE: According to your comment, the values of opacity and linear expansion have been added (P. 7, L. 117-118, P. 9, L. 145-146).

7. *The evaluation of the fluorescence image quality is described in a qualitative way. Why not quantifying the contrast values by taking a cross section and calculating the contrast?*

RESPONSE: Thank you for your suggestion. Accordingly, we have evaluated the fluorescence image quality in a qualitative way. The contrast value you suggested is very useful, however, we have quantified the shape characteristics of the histogram, *VAR* so that we can evaluate the entire image (P. 12-13, L. 211-224, P. 16, 279-283). Meanwhile, we have applied the contrast value to evaluate the transparency of cleared noodles (P. 7-8, L. 121-131).

8. *The 2 previous remarks can be generalized. All the results need to be described in the text in a more quantitative way. This includes the depth of imaging (cfr. 'a longer excitation wavelength enables deeper imaging') and the applied laser powers.*

RESPONSE: Thank you for your comment. We had described the relevant sentence as a general theory in the original manuscript, but as you mentioned, we also needed a quantitative description. However, unfortunately, the two-photon excitation microscope we used does not have the confocal apparatus for one-photon excitation, so the relevant experiment cannot be performed. Thus, the sentence has been deleted. Instead, to provide quantitative data for depth, we have added the *VAR* values depending on the depth, as mentioned above. Moreover, the imaging conditions, including the excitation wavelength and laser power, have been summarised in Supplementary Table 3.

9. *In the conclusion it is mentioned that the findings of this paper can be seen as an alternative for the currently widespread imaging methods for viewing the gluten structure in wheat dough. Can you benchmark your results with the state-of-the-art results? Also with respect to the two researched spectroscopic systems I think it is worth to add a conclusion comparing the performances of both methods for the gluten structure visualization.*

RESPONSE: The main purpose of the protein network analysis, which has been added in response to your comment, is to evaluate the swelling, however, this analysis can also compare the imaging performance for commonly used method and the SoROCS-based method. Moreover, to demonstrate the advantage of SoROCS-based imaging method, we have added an analysis of the volume distribution of gluten, which was difficult to achieve with existing imaging methods (P. 15-16, L. 263-272).

10. *It is reported that the measurements were repeated 20 times. Please elaborate if these were 20 independently prepared samples, if these were 20 samples taken from the same preparation step or if these are 20 measurements made on the same sample. The same holds*

for n=3 mentioned in the caption of figure 1.

RESPONSE: We apologize for the inaccuracies in the statements regarding the repetition of the experiment. We have since included a description for the number of sample preparations and experiments performed to all relevant figure captions (P. 33-34, L. 553-554, P. 35, L. 562, P. 36, L. 570-573, P. 39, L. 604, P. 42, L. 620-621, P. 44, L. 637-638).

11. *It is hard to visually observe a difference for the images shown in the extended data figure 2.*

RESPONSE: According to your comment, we have changed to image sets with the same processing for each set to improve the visibility (P. 37, Fig. 3a-e, g-k).

12. *Extended data figure 3 shows a high similarity with Figure 3. What is the added value?*

RESPONSE: We displayed the same dataset in different ways so that we can focus on the surface or center of the three-dimensional structure of gluten, however, these images only offer minimal information. We have, therefore, in response to your comment, deleted the figures.

Dear Reviewer #2

We sincerely thank you for your valuable comments regarding our manuscript submitted to *Nature Communications* (NCOMMS-20-22137-T). We also appreciate the time and effort you have dedicated to providing insightful feedback on ways to strengthen our paper. Revisions have been made according to your comments, which are indicated in **red text throughout the revised manuscript**.

Responses to the Comments (in *blue italics* the reviewer's comment):

Ogawa and Matsuura developed an optical clearing reagent (SoROCS) to render whole-noodle imaging of wheat. SoROCS enabled 3D imaging of gluten structure in the intact noodles. The manuscript is well written and clearly describes the experimental procedure. But I could not understand the novelty and importance of this method, while my field of expertise is plant cell biology and imaging.

RESPONSE: We have incorporated changes that reflect the detailed suggestions you have graciously provided. We also hope that our edits and the responses we provide below satisfactorily address all of the issues and concerns you have noted.

- 1. The authors should introduce previous research on gluten of the field and properly demonstrate the superiority of the methods they have developed by showing the data. What methods have been used to analyze gluten structure? Electron microscopy? How is it different from that? Are there any specific novelties that the newly developed methods will reveal? The authors should also provide data to show its novelties.*

RESPONSE: In response to your comment, we have added descriptions regarding the previous research on the gluten imaging (P. 4, L. 58-63). The most novel aspect of the method we developed is that it enables three-dimensional quantification. To demonstrate this, we have added an analysis for the volume distribution, which was difficult to achieve with existing imaging methods (P. 15-16, L. 263-272). We believe that such quantitative analysis results will lead to improved quality of wheat-flour foods.

I have outlined the major concerns in detail below:

Major concerns:

- 2. What does "intact" mean? The authors only show the data after swelling, but is there any evidence demonstrated that it is "intact"? What rationale do the authors have that sodium*

salicylate treatment keeps the noodle intact without fixation?

RESPONSE: In response to your comment, we have added a fixation step with paraformaldehyde before clearing (P. 24, L. 411-413). As a result, we found a very good additional advantage that swelling caused by clearing can be suppressed. We sincerely appreciate your suggestion. Although the noodle samples have been fixed, the word ‘intact’ has been deleted throughout the revised manuscript.

3. *P4, L16-17; What kind of serendipity was it? It would be helpful for the reader to have a specific explanation of why the authors tried sodium salicylate.*

RESPONSE: According to your comment, the sentences have been added (P. 5, L. 77-79).

4. *Fig. 1 or P6, L7-9; The superiority of SoROCS cannot be determined without comparison with other transparency reagents.*

RESPONSE: In response to your comment, we have performed a comparison with the previous clearing reagent “ClearSee”. ClearSee made the noodles transparent to some extent, however, the urea contained in the ClearSee seems to have destroyed the gluten structure, as a similar phenomenon was reported in previous studies, even though the noodles were fixed with paraformaldehyde (P. 7-8, L. 109-137).

5. *P7, L13-17; The authors claim to have seen the structure of gluten, but is gluten specifically stained? Is it possible to confirm the specificity of visualized the gluten by other methods?*

RESPONSE: In response to your comment, to confirm the staining specificity of gluten by Thiolite Green (TG), we have performed co-staining with immunochemical staining and TG staining. The plot profile analysis on a merged image showed that the tops of the peaks overlapped with each other, indicating that TG can capture the gluten structures in the noodle (Supplementary information P. 8, Supplementary Fig. 3).

6. *Since the sample is swollen, the authors should also indicate the original scale. "Expansion Microscopy (Chen et al. 2015, Science)" is a good example of how to show the scale for both original and swollen.*

RESPONSE: As you mentioned, examining swelling artifacts is important, and the article "Expansion Microscopy (Chen et al. 2015, Science)" provides a good model method. However, since the noodles are opaque and starch particles are attached to the surface of the noodles, it is difficult to compare the structure of gluten before and

after the clearing treatment. Alternatively, we have performed the protein network analysis to examine the effect of expansion on gluten structures. We found that swelling does not alter the structural or morphological network attributes of the gluten (P. 9-10, L. 151-179).

7. *P8, L17-P9, L5; I didn't understand the need for deep or whole imaging. The authors said that "when comparing the xy images at several depths, they show almost the same structure". If so, whole-imaging is no need?*

RESPONSE: In noodles without salt, a relatively similar honeycomb-shaped network structure is observed, however, in noodles with salt, gluten has a sparse large clump structure; therefore, three-dimensional observation is indispensable. As you correctly pointed out, the relevant text has been deleted as it may be misleading. Instead, to demonstrate the advantages of whole imaging with SoROCS, as mentioned above, we have performed volume distribution analysis (P. 15, L. 263-272).

8. *P8 L4-L5, P9 L5, P9 L18-P10 1; "TG and AF may render more sharply", almost same structure", and "the overall structure became hazy" are "too" subjective. The authors should provide quantitative data from these images.*

RESPONSE: The first of the three sentences you pointed out have been deleted during this revision. Regarding the second point, we have deleted the associated text that you noted in your comment above. For the third point, according to your comment, we have added the percentage of gluten by binarisation (P. 14, L. 236-240).

9. *P9 L7-8; the authors claimed that "this is the first report that renders the 3D structure of gluten, especially in millimetre-scale sizes and submicron resolution.", but which data indicates submicron resolution?*

RESPONSE: The resolution was theoretically calculated from the excitation wavelength and the NA of the objective lens in the original manuscript. However, in response to your comment, we have measured the FWHM from the point-spread function using a fluorescent sphere with a diameter of 0.49 μm and confirmed that the resolution at a depth of 1 mm from the surface of the noodles was 0.51 μm (P. 11-12, L. 191-204).

10. *P19 L18 and Fig 4; The authors should use for the same objective lens or same*

magnification to compare between two-photon and confocal microscopy. Why did the authors choose a 10x objective lens?

RESPONSE: The 25× objective lens dedicated to the upright 2PEM cannot be attached to the inverted CLSM used in this experiment. However, according to your comment, the objective lens used has been changed from 10× to 20×, which provides a similar magnification (P. 28, L. 472-473, P. 44, Fig. 7).

11. *Fig 4; I could not seed the fluorescent signals of the samples near the glass surface, is that a bumpy noodle? It doesn't look that bumpy in the other figures.*

RESPONSE: According to your previous comment, we have remeasured with a 20× objective lens. Similar to the image observed by 2PEM shown in Fig. 4, an image with a smooth surface has been acquired (P. 44, Fig. 7).

Minor concerns:

12. *P16 L4, How many degrees for heating to melt?*

RESPONSE: According to your comment, we have added the heating temperature of 70 °C (P. 23, L.383).

13. *P16 L7-L10; please provide the stock concentration of each dye in DMSO.*

RESPONSE: According to your comment, we have provided the stock concentration (P. 22, L. 388-389).

14. *P9 L1-L2 and P17 L5; The authors said that "SoROCS caused an approximately 2.75-fold linear expansion" and the thickness of noodles was 0.78 mm, but Fig2c showed the thickness was over 2.5 mm. What is this difference?*

RESPONSE: As you pointed out, the calculation results did not match. This may be due to the noodles being tender and, therefore, slightly crushed when measured with calipers. Noodle thickness values may contain measurement errors that can be confusing to the reader; therefore, we have retained the description of the die size (P. 24, L. 398-399), however, the thickness of noodles have been deleted in the revised manuscript.

15. *Extended Data Fig.2a-e, g-k; It's too dark to observe the staining.*

RESPONSE: According to your comment, we have revised the image sets with the same processing for each set to improve the visibility (P. 37, Fig. 3a-e, g-k).

Dear Dr Kontogiorgos (Reviewer #3)

We sincerely thank you for your valuable comments regarding our manuscript submitted to *Nature Communications* (NCOMMS-20-22137-T). We also appreciate the time and effort you have dedicated to providing insightful feedback on ways to strengthen our paper. Revisions have been made according to your comments, which are indicated in **red text throughout the revised manuscript**.

Responses to the Comments (in *blue italics* the reviewer's comment):

The manuscript by Ogawa and Matsumura reports on a new methodology to visualise gluten network in the presence of starch. To allow this, they have established a clearing reagent and new imaging conditions. As a result, the manuscript is more akin to a new experimental protocol. The manuscript is well written and presents new information and has, in my opinion, good chances to be accepted by the relevant research community, as the technique is relatively simple and standard equipment (e.g., CLSM) may be used to carry out the visualisation. In addition, it may help other researchers to establish structure-function relationships in samples that are otherwise difficult to visualise (e.g., bread dough or laminated dough formulations).

RESPONSE: We are very grateful to the reviewer for his very positive opinion about our manuscript. We have incorporated changes that reflect the detailed suggestions you have graciously provided. We also hope that our edits and the responses we provide below satisfactorily address all of the issues and concerns you have noted.

I have a few points that if they are addressed, in my opinion, may help to improve the presentation of the work:

1. *1) The length scale that the structure is observed is in the range of ~100 μm . This needs to be clearly mentioned in the manuscript and contrasted with other techniques that provide "deeper" visualisation of the structure. For example, what is the structure of EACH strand that is observed in Figure 2, 3 and 4? Unfortunately, it is not possible to obtain this information with this technique.*

RESPONSE: We have revised the text according to your suggestions (P. 12, L. 208-211).

2. *2) The present visualisation method has not been compared with other previously published work. How does the new technique compare with already published work? Has new*

structural information been revealed or it is just an additional experimental technique in the toolkit of the researchers?

RESPONSE: According to your comment, we have conducted a comparative test with the previous clearing reagents (P. 7-8, L. 109-137). Moreover, we have added the protein network analysis to compare the imaging performance of commonly used methods and the SoROCS-based method (P. 9-10, L. 151-179). We believe that the SoROCS-based imaging method will not only replace the currently widespread imaging methods but will also enable new quantitative analysis. To demonstrate this advantage, we have included a volumetric analysis (P. 15-16, L. 263-272).

3. *3) There is a section with advantages of SoROCS. A section with disadvantages and limitations also needs to be present, as there is no such technique without its disadvantages.*

RESPONSE: According to your comment, we have modified the advantages section to describe both the advantages and disadvantages (P. 16, L. 287, P. 17, L. 296-300).

Minor comments

4. *Lines 112-113: Gelatinisation cannot be achieved without heating. Do you mean “swelling”? Please, rephrase.*

RESPONSE: Thank you for your comment. We have revised “gelatinisation” to “swelling of starch” (P. 9, L. 150).

5. *Lines 181-185: This information is well documented and routinely observed in the literature. Please, cite key past work to complete the discussion of this part.*

RESPONSE: As you pointed out, this information is well documented; therefore, we have added the citation for the latest article (P. 15, L. 259-261, P. 21, L. 376-378).

6. *Please, explain the purpose of Triton in your clearing reagent.*

RESPONSE: According to your comment, we have included an explanation for the purpose of Triton X-100 (P. 6, L. 100).

7. *Figure 3. What is the size of the scalebar?*

RESPONSE: In response to your comment, we have added the size of the scale bars (P. 41, L. 615).

8. *Title needs to be revised as it is not informative. A “giant protein” should be replaced by “gluten”.*

RESPONSE: Thank you for your suggestion. According to your comment, we have revised the title (P. 1, L. 1).

REVIEWER COMMENTS

[Editor: Reviewer #1 states in Remark to Editor section that (s)he is satisfied with the revision.]

Reviewer #2 (Remarks to the Author):

In this revised version, the authors have much improved their manuscript by addressing all my comments.

I still have very few comments on some points.

p8 line 133-134, SeeDB paper only mentioned denaturation of cellular proteins by urea but did not show any results.

Fig. 7, the authors showed the CLSM image up to a depth of 600 μm but I think SoROCS-treated samples could be observed deeper by using a lens with a longer working distance. It would be good to describe the working distance of the lens to emphasize the usefulness of SoROCS in CLSM.

Reviewer #3 (Remarks to the Author):

The authors have addressed all my comments and in my opinion the manuscript can now be accepted for publication.

Dear Reviewer #1

RESPONSE: We appreciate your time and effort in reviewing our manuscript.

Dear Reviewer #2

We sincerely thank you for your valuable comments regarding our manuscript submitted to *Nature Communications* (NCOMMS-20-22137-A). We also appreciate the time and effort you have dedicated to providing insightful feedback on ways to strengthen our paper. Revisions have been made according to your comments, which are indicated in **red text throughout the revised manuscript**.

Responses to the Comments (*blue italics*, reviewer's comment):

In this revised version, the authors have much improved their manuscript by addressing all my comments. I still have very few comments on some points.

1. *p8 line 133-134, SeeDB paper only mentioned denaturation of cellular proteins by urea but did not show any results.*

RESPONSE: According to your comment, the sentences have been modified (P. 8, L. 142-144).

2. *Fig. 7, the authors showed the CLSM image up to a depth of 600 μm but I think SoROCS-treated samples could be observed deeper by using a lens with a longer working distance. It would be good to describe the working distance of the lens to emphasize the usefulness of SoROCS in CLSM.*

RESPONSE: Thank you for your comment. We have added a description about the working distance (P. 17, L. 289-290).

Dear Dr Kontogiorgos (Reviewer #3)

Responses to the Comments (*blue italics*, reviewer's comment):

The authors have addressed all my comments and in my opinion the manuscript can now be accepted for publication.

RESPONSE: We appreciate the reviewer's positive comments on our manuscript.

REVIEWERS' COMMENTS

Reviewer #2 (Remarks to the Author):

The authors have satisfactorily addressed all my comments.

Dear Reviewer #2

Response to the Comment (*blue italics*, reviewer's comment):

The authors have satisfactorily addressed all my comments.

RESPONSE: We appreciate your time and effort in reviewing our manuscript.